# Single-cell mass cytometry and transcriptome profiling reveal the impact of graphene on human immune cells

Marco Orecchioni[1], Davide Bedognetti [2], Leon Newman[3], Claudia Fuoco[4], Filomena Spada[4], Wouter Hendrickx [2], Francesco M. Marincola [5,8], Francesco Sgarrella[1], Artur Filipe Rodrigues [3], Cécilia Ménard-Moyon[6], Gianni Cesareni [4], Kostas Kostarelos [3], Alberto Bianco [6] & Lucia G Delogu [1,7]

Understanding the biomolecular interactions between graphene and human immune cells is a prerequisite for its utilization as a diagnostic or therapeutic tool. To characterize the complex interactions between graphene and immune cells, we propose an integrative analytical pipeline encompassing the evaluation of molecular and cellular parameters. Herein, we use single-cell mass cytometry to dissect the effects of graphene oxide (GO) and GO functionalized with amino groups ($GONH_2$) on 15 immune cell populations, interrogating 30 markers at the single-cell level. Next, the integration of single-cell mass cytometry with genome-wide transcriptome analysis shows that the amine groups reduce the perturbations caused by GO on cell metabolism and increase biocompatibility. Moreover, $GONH_2$ polarizes T-cell and monocyte activation toward a T helper-1/M1 immune response. This study describes an innovative approach for the analysis of the effects of nanomaterials on distinct immune cells, laying the foundation for the incorporation of single-cell mass cytometry on the experimental pipeline.

[1] Department of Chemistry and Pharmacy University of Sassari, Sassari 07100, Italy. [2] Immunology and Therapy Section, Division of Translational Medicine, Sidra Medical and Research Center, Doha, Qatar. [3] Nanomedicine Laboratory, Faculty of Biology, Medicine & Health, and National Graphene Institute University of Manchester, Manchester M13 9PL, UK. [4] Department of Biology, University of Rome Tor Vergata, Rome 00133, Italy. [5] Office of the Chief Research Officer, Sidra Medical and Research Center, Doha, Qatar. [6] CNRS, Institut de Biologie Moléculaire et Cellulaire, Laboratoire d'Immunopathologie et Chimie Thérapeutique, Strasbourg 67 084, France. [7] Max Bergmann Center of Biomaterials and Institute for Materials Science, Dresden University of Technology, Dresden 01069, Germany. [8] Present address: Abbvie Corporation, Redwood City, CA 94063, USA. Correspondence and requests for materials should be addressed to K.K. (email: kostas.kostarelos@manchester.ac.uk) or to A.B. (email: a.bianco@ibmc-cnrs.unistra.fr) or to L.D. (email: lgdelogu@uniss.it)

The development of nanomaterials for medical and diagnostic applications[1] is one of the most promising frontiers of nanotechnology. Graphene, a single layer of hexagonally arranged carbon atoms, and graphene oxide (GO), the oxidized form of graphene, are carbon nanomaterials of extraordinary physicochemical properties and a biocompatible profile that enables their utilization in biomedical applications[2–4]. However, the impact of GO exposure on the immune system remains unclear[5–7]. Differences among reports could be attributed to the variability in the physicochemical characteristics of materials used in terms of lateral dimensions, surface functionalization, and chemical purity and deserves further investigation[8–10].

GO can be rich in functional groups such as epoxy and hydroxyl groups, which facilitate its surface modifications increasing its biocompatibility. GO has been investigated in a continuously growing number of medical applications[11, 12]. However, the main limitation in using GO in nanomedicine is its biocompatibility. As such, the evaluation of the immune perturbations induced by nanoparticles is an essential prerequisite.

On the other hand, specific toxic effects of graphene-based materials on cancer cells support its use in nanomedicine[13, 14], for example, as an inhibitor of cancer cell metastasis[15] or as a passive tumor cell killer in leukemia[16].

As mentioned above, the effects played by physicochemical characteristics of nanomaterials in terms of lateral dimension, functionalization, and purity are still under discussion. In this context, the chemical modifications of graphene can play a role in the impact of these nanoparticles on the immune system[8]. It was already reported that functionalization can reduce the toxicity by changing the ability of graphene to modulate the immune response[6]. Similarly, the cyto- and genotoxicity of reduced GO (rGO) sheets on human mesenchymal stem cells were found to depend on the lateral dimensions of the materials, ultra-small sheets being more toxic[17, 18]. Studies have also shown that the aspect ratio of the graphene sheets is an important factor to consider. For instance, rGO affects cell viability only at very high concentration (i.e., $100 \, \mu g \, ml^{-1}$), while single-layer GO nanoribbons display significant cytotoxic effects at $10 \, \mu g \, ml^{-1}$[19]. Moreover, a direct impact on the antibacterial activity or on reproduction capability of mice influenced by the aspect ratio of GO has been reported[19–21]. The possibility to rationally design graphene materials with different physicochemical characteristics could expand further their application in medicine[22].

The understanding of the complex interactions between nanoparticles and immune cells is hindered by insufficient implementation of high-throughput, deep phenotyping technologies in the field[23–26]. The immune system is a sophisticated machine meant to protect the body against injury, pathogens, or tumors. Its dysfunction can induce pathologies such as autoimmune diseases, allergies, and cancer[27, 28]. Revealing the interactions of different GOs with this complex system still remains a challenge.

Such a study should include tools that permit the multiplex analysis of cell type, activation status, and release of soluble mediators with stimulatory and inhibitory properties[28, 29].

Flow cytometry has been primarily used to address single-cell behavior. Recently, a tool employing mass spectrometry has been developed to leverage the precision of flow cytometry analysis. The combination of the two techniques, termed single-cell mass cytometry (CyTOF), allows the simultaneous measurement of more than 40 cellular parameters at single-cell resolution with over 100 available detection channels[30, 31].

Compared to fluorescence-based cytometry, mass cytometry employs element-tagged probes that enable the discrimination of elements according to their mass/charge ratio ($m/z$), with minimal overlap and background cellular signal. All these attributes simplify the large panel experimental design, thus uniquely enabling high-dimensional cytometry experiments that would not be possible otherwise[30, 32–34].

In the present work, we demonstrate the use of single-cell mass cytometry together with whole-transcriptomic analysis to dissect the immunological effects of nanomaterials on individual cells. Our results emphasize the importance of the functionalization on enhancing the biocompatibility of GO-based nanomaterials. Notably, only the amino-functionalized GO was able to induce a specific monocytoid dendritic cell (mDCs) and monocyte activation skewed toward a T helper (Th)-1/M1 response. These findings are starting points for the development of nanoscale platforms in medicine as immunotherapeutics, vaccine carrier, or nanoadjuvant tools.

## Results

**Graphene synthesis, functionalization, and characterization.** Thin GO flakes (single to few graphene layers) and GO surface-functionalized with amino groups ($GONH_2$)[35] via epoxide ring opening, using triethyleneglycol (TEG) diamine, were investigated (see Methods section). We have previously shown that the epoxide ring opening reaction is a versatile strategy to functionalize GO in a controlled manner. This reaction targets the epoxide groups, without causing reduction of the starting GO material[35]. The detailed physicochemical characterization of both GO and $GONH_2$ is reported in the Supporting Information (Supplementary Fig. 1). Briefly, the 2D material morphology was studied by both TEM and AFM (Supplementary Fig. 1a–d). These techniques indicated that the lateral sheet dimensions of both GO and $GONH_2$ ranged between 50 nm and 1 μm. The vast majority of the flakes, both for GO and $GONH_2$, had lateral dimensions smaller than 300 nm (73% and 62%, respectively, according to TEM measurements). The height distributions obtained by AFM revealed that GO sheets had thicknesses corresponding to single and few (2–3) layers. The $GONH_2$ sheets were ~3 times thicker than the non-functionalized GO. The increased thickness of graphene-based materials following functionalization has been previously reported and is attributed to the reaction processing and the presence of functional groups on the sheet surface[36, 37].

Raman spectroscopy evidenced the presence of the characteristic D and G bands (1330 and 1595 $cm^{-1}$, respectively) in both GO materials, confirming their graphenaceous structure. Furthermore, we observed that both GO and $GONH_2$ exhibited a 2D band that was of low intensity and of broad linewidth. This correlated with the oxidation and exfoliation of graphite to GO in the modified Hummers' method, due to increased defects in the graphene sheets[38, 39]. Therefore, the analysis of this peak is not a reliable indicator to draw conclusions regarding the presence of single-layer GO, even though it is possible in the case of graphene[40] and rGO[41]. Nonetheless, we examined the I(D)/I(G) ratio, a commonly used parameter to assess disorder[42]. When comparing GO and $GONH_2$, the I(D)/I(G) ratio did not increase significantly, since the epoxy ring opening reaction conditions used for amination were shown to not add further defects to the GO surface[35, 38]. It was reported that a strong reduction process results in an increased I(D)/I(G) ratio due to the predominance of small $sp^2$ carbon domains in the graphene lattice[43]. These results are consistent with the maintenance of the oxidation degree of GO after functionalization, which was demonstrated in a previous study by XPS analysis of the C-O binding energy peak, before and after functionalization with TEG diamine[30]. We found that the O/C ratio decreased from 0.44 to 0.38 after GO covalent modification. It was reported that ethylenediamine is able to reduce GO, but the mechanism involves the formation of a five-membered ring that is not possible using TEG diamine[44].

Stronger agents like hydrazine or plant extracts are necessary to achieve an efficient reduction of GO[44, 45]. Amino functionalization was confirmed by FT-IR spectroscopy, which showed a clear difference between GO and GONH$_2$. Importantly, the presence of an extra band in the 1260–1330 cm$^{-1}$ range in GONH$_2$ compared to the GO samples can be explained by the amine C-N stretching and C-H bending. Furthermore, the presence of a new band around 2900 cm$^{-1}$, indicative of the presence of the aliphatic C-H stretching, supported the successful functionalization of GO by epoxide ring opening due to the presence of the TEG chain. These results provided evidence of the successful synthesis of GO and GONH$_2$ studied in our experiments[35, 46].

**Dissecting the immunological impact of graphene with CyTOF**. We used single-cell mass cytometry to analyze simultaneously 30 markers discriminating distinct subpopulations of peripheral blood mononuclear cells (PBMCs), in order to understand the response to nanomaterial exposure. CyTOF analysis allowed us to check the differential viabilities of 15 immune cell subpopulations exposed to GO and GONH$_2$ for 24 h. A concentration of 50 µg ml$^{-1}$ was chosen for these experiments because it was identified as an appropriate concentration for different GO-based biomedical applications[9, 23, 47]. One qualitative difference between flow cytometry and mass cytometry is the absence in the latter of the spectral overlap that complicates the analysis of fluorescence data. Another advantage is the absence of cell-dependent background signals in the mass cytometry data[48].

Immune cell populations are identified according to the expression profile of cluster of differentiation (CD) markers present on the cell surface. When immune cells go through different stages of maturation and differentiation, the CD marker profiles change. Mass cytometry with its high dimensionality is an ideal approach to simultaneously characterize several cell markers. CyTOF could analyze the effect of GO and GONH$_2$ on a wide variety of immune cell populations, determining also different maturation and activation stages. To reduce the dimensionality of the data set, we used SPADE (spanning tree progression analysis of density-normalized events) clustering algorithm (Fig. 1a), as reported by Bendall et al.[48] To construct the SPADE tree, we used 11 cell surface markers in treated and untreated healthy human PBMCs to identify the major immune cell populations. Sixteen additional markers were acquired. Among them, five extracellular markers were used to better define cell subpopulations. The remaining 11 intracellular makers were employed for the cytokine detection, and were excluded from the tree construction. Each node in the two dimensional representation describes an n-dimensional boundary encompassing a population of phenotypically similar cells. The size of each node in the tree is proportional to the number of cells within each population. Node color is scaled to the median intensity of marker expression. The approach uses a minimum-spanning tree algorithm, in which each node of cells is connected to its most related node as a means to convey the relationships between the cell clusters. As a result, the 15 manually assigned populations were segregated in 200 nodes of distinct but logically interconnected populations. These trees provide a convenient approach to map complex n-dimensional relationships into a representative 2D structure[48].

However, it is well known that the SPADE algorithm is inherently stochastic and the estimate of the cell populations can differ in across repeats of the analyses. This limitation of SPADE is also supported by the continuous development of efficient algorithms[49]. To corroborate the robustness of our conclusions, the SPADE analysis was performed three times.

As reported in Supplementary Table 1, the event counts of the main immune subpopulations are similar in the different algorithm runs, confirming the robustness of the SPADE data analysis.

In this analysis, cisplatin (CIS) was used as marker for viability[50]. CIS is a molecule able to enter into the late apoptotic and necrotic cells that have lost membrane integrity. The SPADE tree clustering shows that GO induced cytotoxicity in all B-cell subpopulations (Fig. 1a). Monocytes and activated Th cells were also affected by the presence of GO. On the other end, the functionalized GONH$_2$ significantly reduced CIS signal in all subpopulations (Fig. 1b), CIS median expression in distinct subpopulations. Moreover, GONH$_2$ was three times more biocompatible in all B-cell populations than non-functionalized GO. This effect was also evident in activated cytotoxic T lymphocytes (CTLs) and Th cells, where GONH$_2$ did not induce high levels of toxicity. Overall, the functionalization of GO enhanced its biocompatibility toward the immune populations analyzed (Fig. 1b) with the exception of natural killer (NK) cells and memory CTLs (see Methods section for gating strategy), in which both GO and GONH$_2$ induced minimal cytotoxicity. These results emphasize the importance of amino functionalization in enhancing the biocompatibility of GO-based nanomaterials. Interestingly, the same type of functionalization used to modify GO was previously found to improve the biocompatibility of other nanomaterials such as carbon nanotubes[24, 25].

**Cytokine analysis on several immune cells with CyTOF**. We further applied CyTOF to understand the functional impact of GO and GONH$_2$ on the immune subpopulations. The heat map visualizes the median expression values of all intracellular markers used for each immune population (Fig. 2a). GO caused a broad, non-cell-specific activation triggering the production of all cytokines analyzed in a variety of cell populations (Fig. 2a), while GONH$_2$ was more specific affecting, for instance, the production of only few cytokines in selected cell subpopulations. Among T cells, GO induced the secretion of interleukin (IL) 2, 4, and 5 by Th and CTLs (Fig. 2b). Conversely, GONH$_2$ selectively induced the production of IL2 by activated T cells and the production of tumor necrosis factorα (TNFα) in several cell subpopulations. Moreover, GONH$_2$ did not affect the synthesis of IL5 by T cells, and only modestly effected that of IL4 (Fig. 2b). IL4/IL5 are markers of Th2 polarization, while TNFα/IL2 indicate Th1 differentiation. Thus, it appears that GONH$_2$ elicits a polarized Th1 immune response and a non-specific Th response. GONH$_2$ tropism for Th1 cytokines was mirrored in B-cells (Fig. 2c). Moreover, GONH$_2$ was able to induce dendritic cell and monocyte activation skewed toward a M1 response, as demonstrated by increased production of classic M1 cytokines such as TNFα, IL6, and the CCR5 ligand CCL4 (MIP1β) (Fig. 2d)[51, 52].

Th2 responses are involved in asthmatic reactions and induction of allergy[53]. Moreover, Th2 responses (sustained by M2 macrophages) favor cancer growth[54]. Conversely, Th1 responses (sustained by M1 macrophages) counteract cancer development[55, 56]. In fact, intratumoral Th1 (but not Th2) signatures have been invariably associated with favorable prognosis and responsiveness to immunotherapy[55, 57–62]. These data are of particular interest for further translational applications of amino-functionalized GOs for possible immunotherapeutic strategies or as a vaccine adjuvant. M1 cytokine production such as IL6, TNFα, and MIP1β after treatment with GO and GONH$_2$ is represented in Fig. 3. Negative controls are reported in Supplementary Fig. 2. We found increased expression of IL6 in monocytes, mDCs (monocytoid dendritic cells), and activated Th cells (red nodes) mediated mainly by GONH$_2$ (Fig. 3a). As

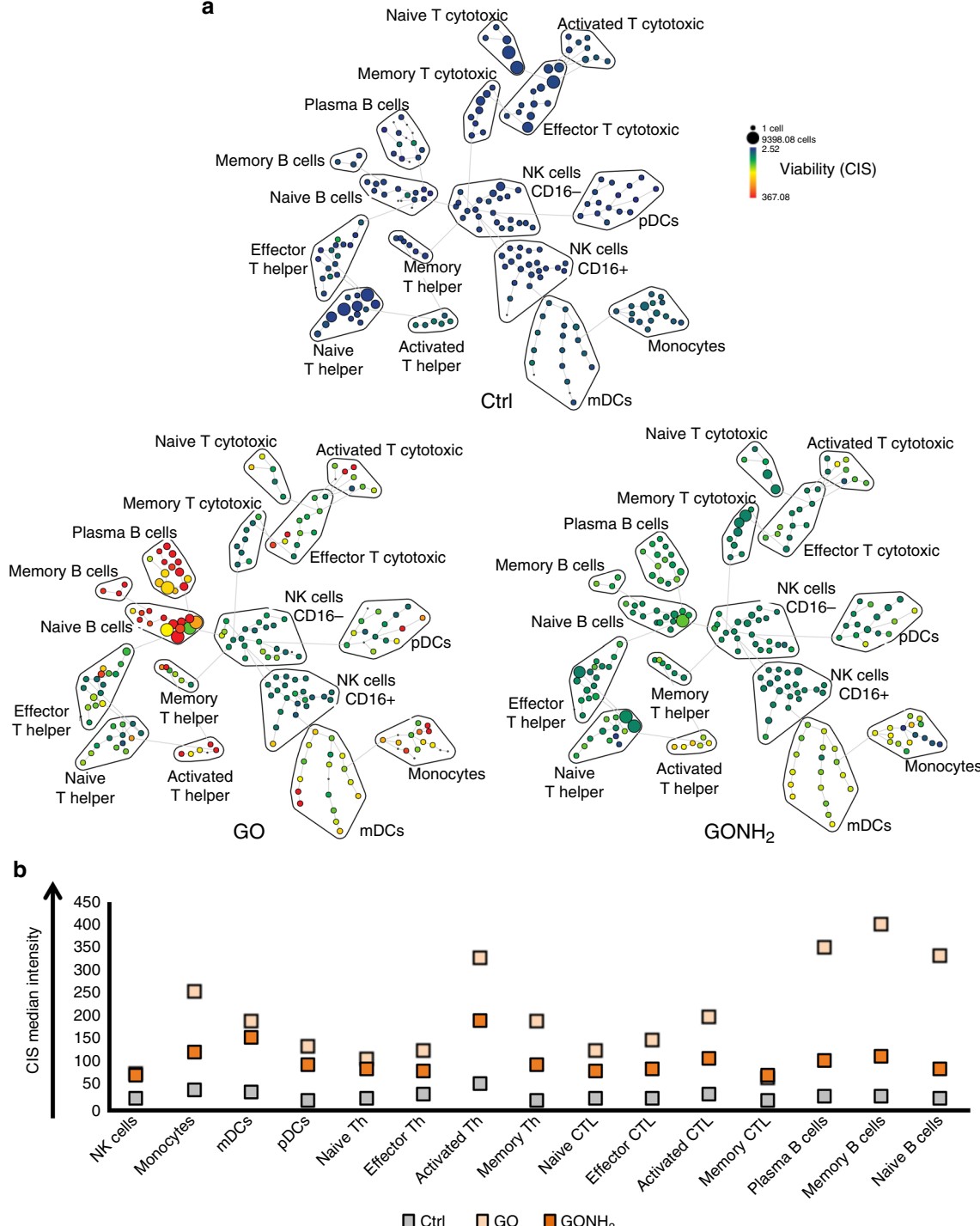

**Fig. 1** Cell viability analysis using CIS reagent with CyTOF. PBMCs were treated with GO and GONH₂ at the concentration of 50 µg ml⁻¹. **a** The SPADE tree plots show the different gated immune cell subpopulations. The size of each cluster in the tree indicates the relative frequency of cells that fall within the dimensional confines of the node boundaries. Node color is scaled to the median intensity of marker expression of the cells within each node, expressed as a percentage of the maximum value in the data set (CIS is shown). **b** The graph reports the CIS median intensity in all subpopulations analyzed. The analysis is made out of three experiments. CTL cytotoxic, T lymphocytes, Th T helper

expected, TNFα secretion was mediated by GONH₂ in monocytes, mDCs, activated CTLs, Th cells, and in NK cells (Fig. 3b). The expression of MIP1β was clearly observable in monocytes, mDCs, and activated Th in response to both GOs. However, as previously mentioned, the median intensity was higher in GONH₂-treated samples. GO, instead, induced MIP1β expression also in B-cell populations (Fig. 3c).

SPADE visualization could give further information on the impact played by GOs on single cells through heterogeneity analysis within nodes. Indeed, not all the cells in the same family display the same cytokine secretion intensity, underlining possible different maturation and/or activation stages. An example is given by TNFα secretion by CD16⁻ NK cells treated with GONH₂ (Fig. 3b), where half of the nodes included did not secrete TNFα.

**Graphene activity evaluation on a single-cell resolution.** Overall, the SPADE data suggest a cross-talk between monocytes/ mDCs and CTLs/Th cells that could sustain a specific cell-mediated immunity, avoiding humoral response and possible hypersensitivity. However, the SPADE visualization fails to preserve the single-cell resolution of the mass cytometry data. For this reason, we applied a second dimensionality reduction method called viSNE, which is a computational approach suitable for the visualization of high-dimensional data with single-cell resolution[63]. By this approach, immune cell phenotypes are

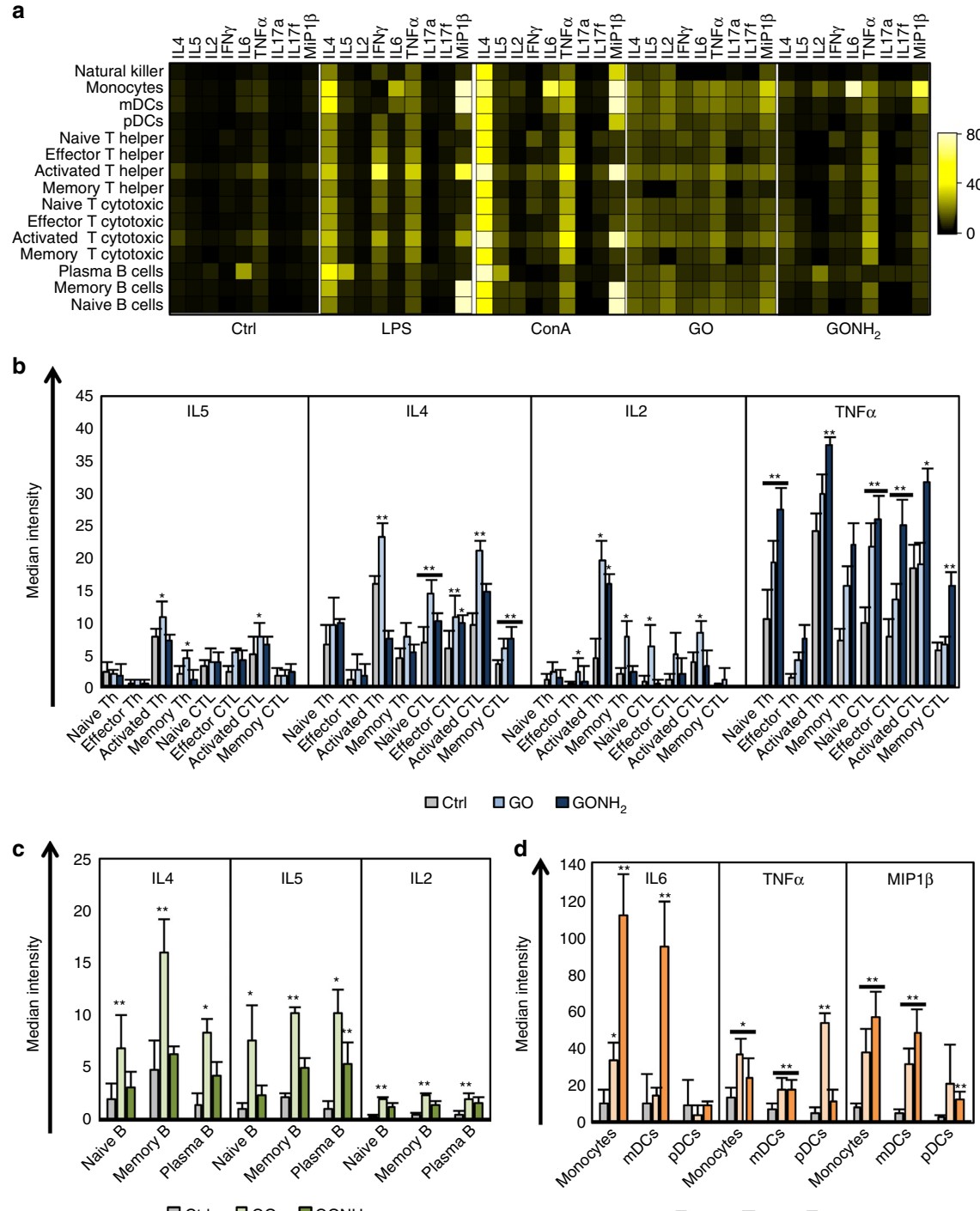

**Fig. 2** Analysis of the immune cell behavior using CyTOF. PBMCs were treated with GO and GONH$_2$ at the concentration of 50 μg ml$^{-1}$ labeled with markers of immune cell lineages and cytokines, detected with CyTOF2. **a** Heat map of median marker expression ratio for gated immune cell populations. Histograms of intracellular cytokine median expression. **b** IL4, IL5, IL2, and TNFα in T-cell subpopulations. **c** IL4, IL5, and IL2 median expression in B-cell subpopulations. **d** IL6, TNFα, and MIP1β median expression in monocytes and mDCs. The analysis is made out of three experiments (*P-value < 0.05, **P-value < 0.01 Statistical analysis performed by one-way ANOVA test between the median expression of each node into the boundaries compared with the control)

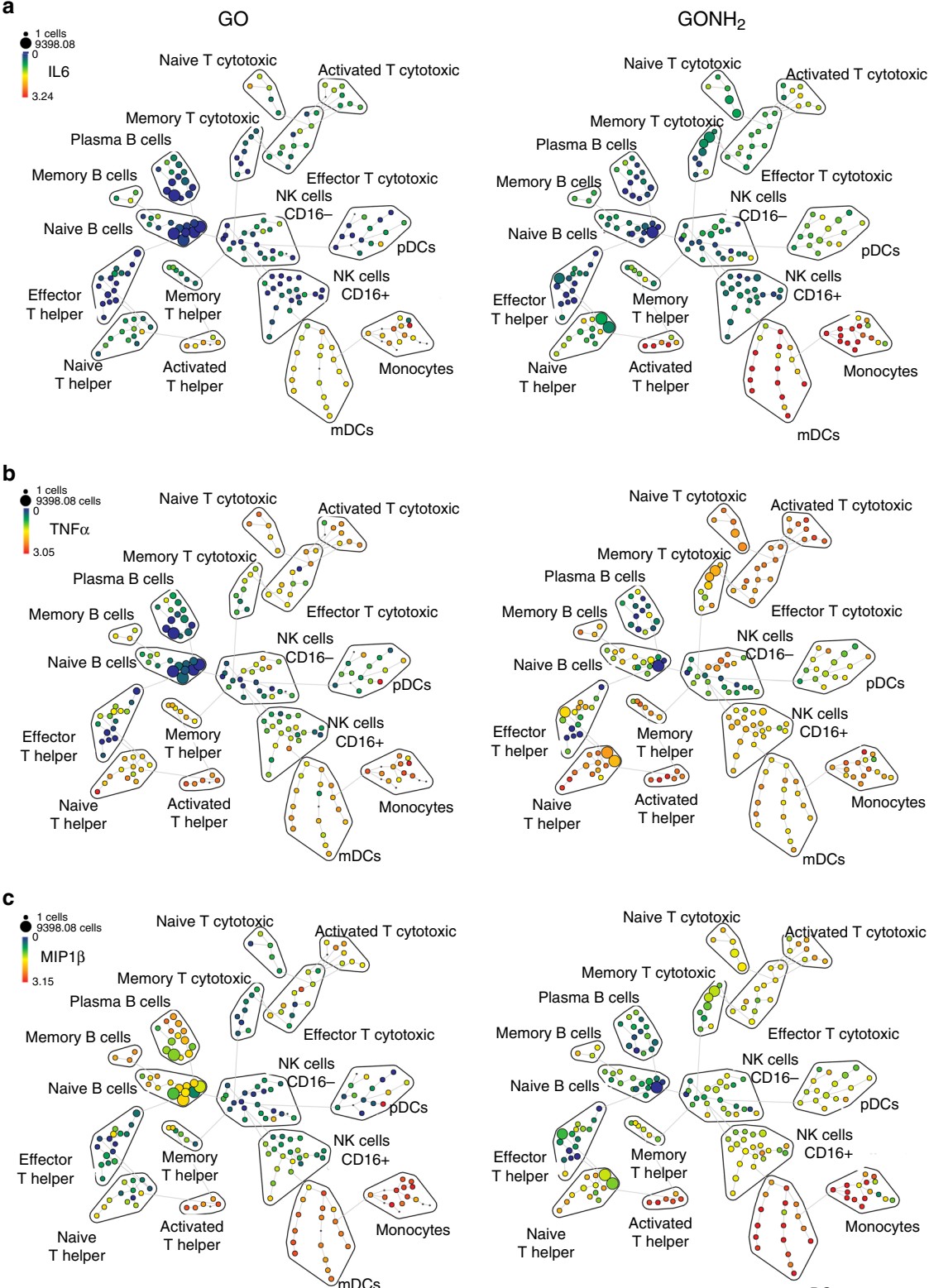

**Fig. 3** Summary of SPADE analysis of significantly secreted cytokines. The tree plots were constructed in the same way of Fig. 1. Node color is scaled to the median intensity of marker expression of the cells within each node, expressed as a percentage of the maximum value in the data set. The spade trees show the median expression intensity of **a** IL6; **b** TNFα, and **c** MIP1β, for GO and GONH₂-treated samples. The analysis is made out of three experiments

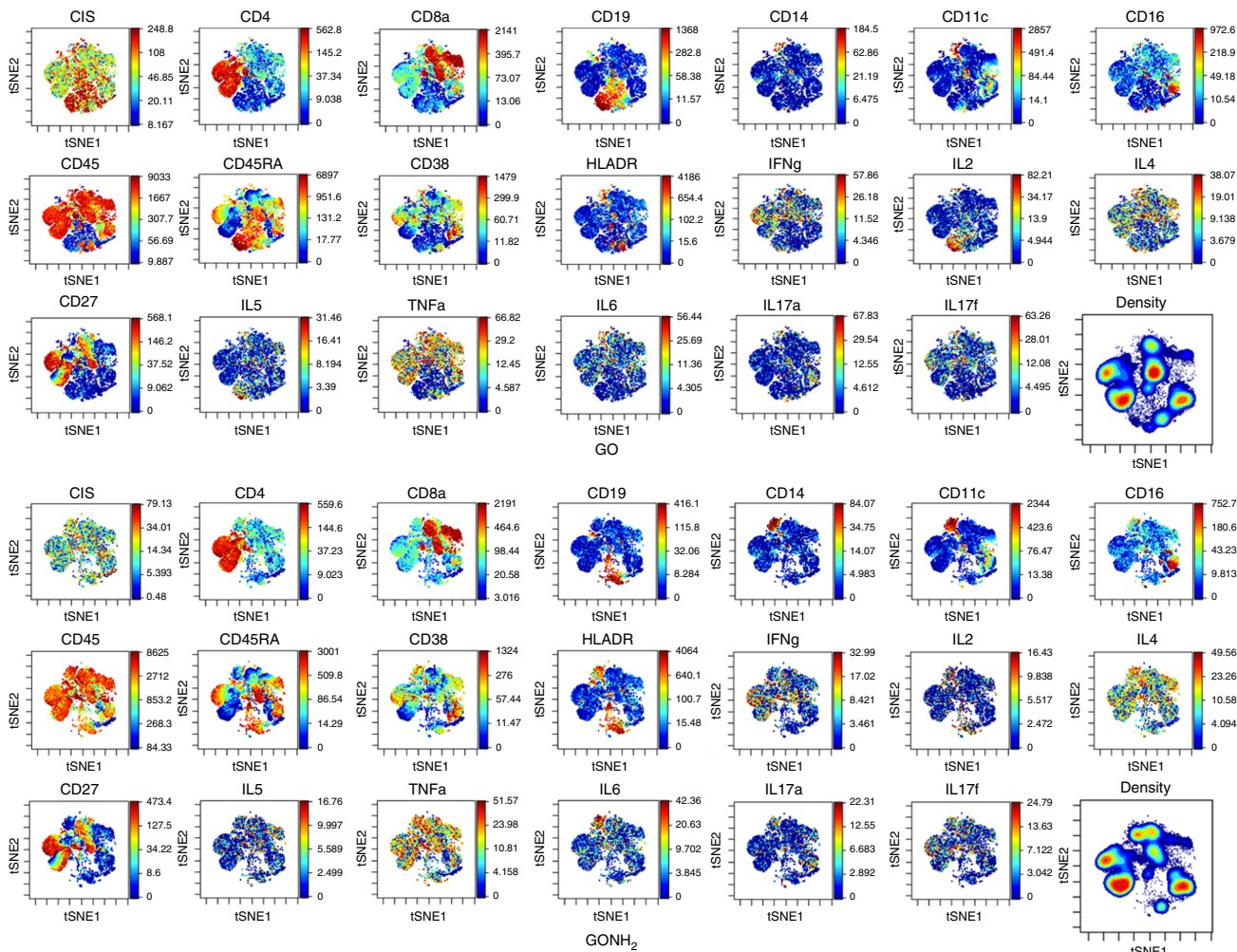

**Fig. 4** Single-cell characterization of GO- and GONH₂-treated PBMCs (viSNE analysis). Plots show the use of viSNE to obtain a comprehensive single-cell view and to distinguish the PBMC subpopulations in the GO- and GONH₂-treated cells. Plots show expression of the 19 proteins, nucleic acid intercalator (CIS), and density measured per cell

projected onto a biaxial plot space according to the similarity of their multidimensional phenotypic expression vector. Thus, viSNE clusters the single-cell events into populations according to the 11 protein expression readouts used in the analysis (Fig. 4). The viSNE analysis accurately identified helper and CTL T cells, B cells, monocytes, and NK cells (Supplementary Figs. 3–6). The naive, memory, and activated T-cell subpopulations and the B-cell subpopulations were also identified (Supplementary Figs. 3–5).

We further exploited the viSNE analysis to investigate the single-cell cytokine profile in response to GO and GONH₂ treatment. This analysis confirmed the subpopulations and cytokine expression profiles obtained by the SPADE approach and supports the main conclusion that the amino functionalization of GO significantly increases cell biocompatibility and polarizes a specific cell activation toward a T helper-1/M1 immune response not affecting the B-cells response. Instead, GO incubation caused an increase in B-cell counts correlating with an increase in IL2 secretion mostly by the plasma B cells and a reduction of monocytes (Supplementary Figs. 5, 6). The single-cell resolution obtained with the viSNE analysis evidenced a heterogeneity in the cytokine expression profile within the same subpopulation (TNFα, IL6, IL5, IL4, and IFNγ) (Fig. 4), revealing a heterogeneous response after GO and GONH₂ treatment.

**Cross confirmation of single-cell analysis.** Since single-cell mass cytometry has not been previously applied to nanomaterials, we corroborated the analysis with several classical techniques to analyze cell apoptosis, necrosis, and activation and cytokine secretion. Analysis of human PBMCs using flow cytometry confirmed the trend observed with the CyTOF experiments. Figure 5a displays the histograms related to apoptosis and necrosis experiments (expression of Annexin V (apoptotic) and PI (necrotic) positive cells after treatment with GO and GONH₂) using the same conditions reported for the CyTOF analysis (P-value < 0.05). Data were also confirmed by a dose-response analysis (5, 25, and 50 μg ml⁻¹) using 7-amino actinomycin D (7AAD) to detect cells with compromised membranes (late apoptotic and necrotic cells). High amounts of necrotic cells were detected, suggesting a possible direct effect of GO on the cell membrane that leads to extensive damage. These findings were in agreement with a previous work in which we disclosed the mask effect of GO[9]. As expected, the functionalization improved the biocompatibility of GO with a reduction of necrotic events, from 42.0 to 24.7% (P-value = 0.045) (Fig. 5a). Similar results are reported in Fig. 5b. Indeed, at the highest concentration used, we found a reduction of necrotic events from 27.3 to 6.7% (P-value = 0.039) in GONH₂-treated samples. The improvement of biocompatibility mediated by GONH₂ was confirmed by hemolysis

analysis in red blood cells (RBCs). Hemolysis is reported to be an undesirable effect mediated by GOs at high concentrations[64]. The release of hemoglobin from damaged RBCs after treatment with increasing doses (5, 25, 50, and 100 µg ml$^{-1}$) of GO and GONH$_2$ was analyzed (Supplementary Fig. 7). The highest concentration of GO was able to induce a significant release of hemoglobin, showing damage to RBCs. On the other hand, the functionalized GONH$_2$ did not damage RBCs at any of the concentrations studied (Supplementary Fig. 7). However, this may be due to the

differences between the two material types in thickness and quality of the suspension, as the GO sheets are dispersed more homogeneously than GONH$_2$.

Further activation analyses were performed by flow cytometry, measuring CD69 and CD25, early and late activation markers, respectively (Fig. 5c). Total PBMCs were treated with both GO and GONH$_2$ at the concentration of 50 µg ml$^{-1}$ for 24 h. GONH$_2$ induced higher (15.03%, P-value 7.89E−05) expression of CD25 compared to GO (8.7%, P-value = 0.009) (Fig. 5c). A similar trend

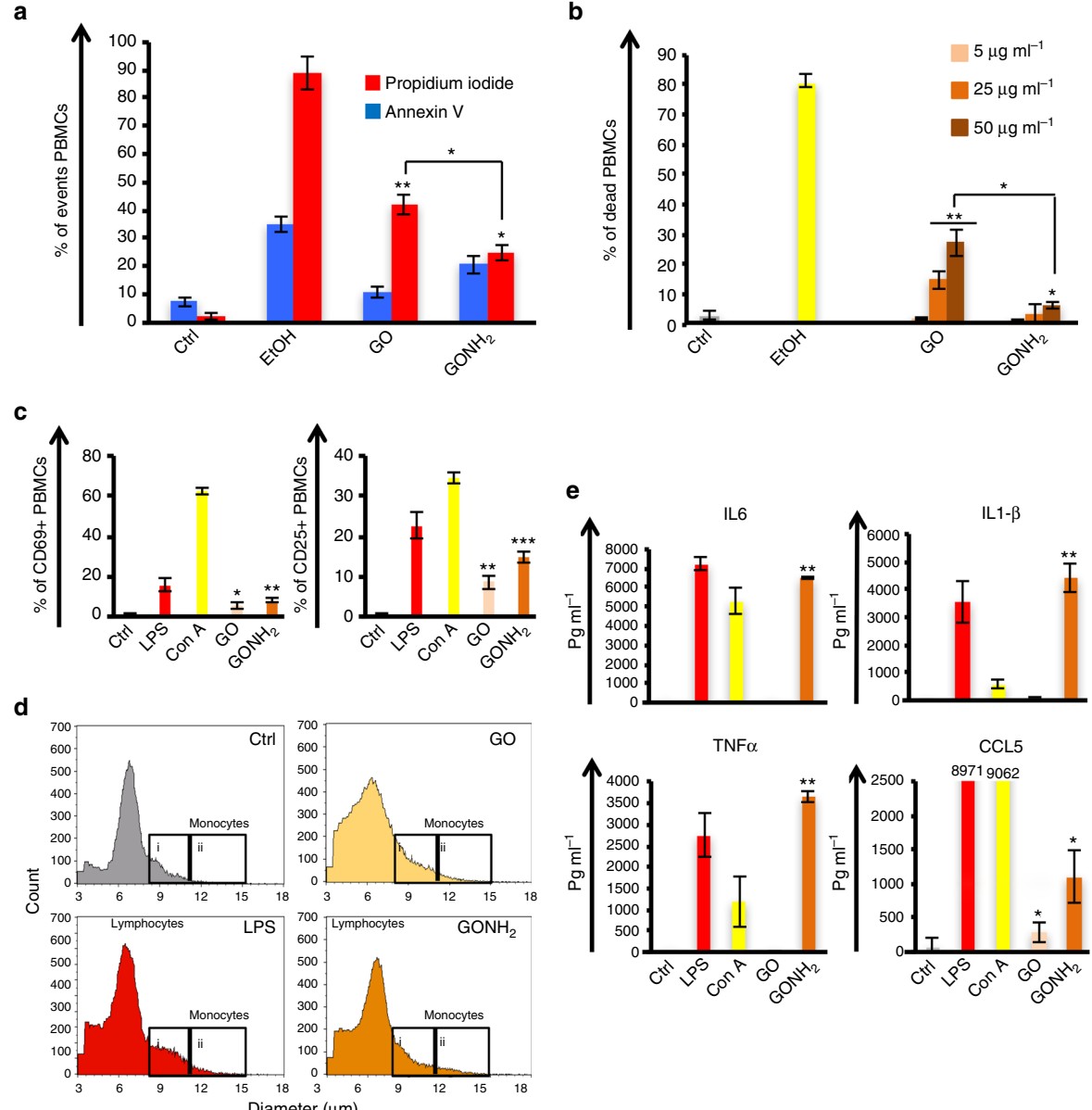

**Fig. 5** Standard cell viability and activation assays in ex vivo PBMCs. PBMCs were incubated with GO and GONH$_2$ at increasing doses (5, 25, and 50 µg ml$^{-1}$) and at a fixed dose (50 µg ml$^{-1}$) or left untreated (Ctrl). Data were analyzed using different stainings by flow cytometry. **a** Necrosis and apoptosis were assessed treating the cells with 50 µg ml$^{-1}$ using propidium iodide and Annexin V staining; ethanol was used as positive control. **b** Percentage of late apoptotic and necrotic cells was assessed by staining with an amine-reactive dye (7AAD) after 24 h of incubation; ethanol was used as a positive control. Experiments were performed at least in triplicate (*P-value < 0.05, **P< 0.01, ***P-value 0.0001). **c** Percentage of CD25 and CD69 cell surface activation marker expression in monocytes (50 µg ml$^{-1}$) (CD14 positive cells) analyzed by flow cytometry. **d** Morphological analysis (count and diameter) of PBMCs using Scepter 2.0 highlighting the monocyte peaks (i), and activated monocytes (ii) with a diameter higher than 11.25 µm. The experiment is reported out of three; **e** Cytokine release was assessed by multiplex ELISA on PBMCs and expressed as pg ml$^{-1}$. Surfactants of cells incubated with GO and GONH$_2$ (µg ml$^{-1}$) were harvested and analyzed by multiplex ELISA. Concanavalin A (ConA, 10 µg ml$^{-1}$) and lipopolysaccharides (LPS 2 µg ml$^{-1}$) were used as positive controls. All the experiments were performed at least in triplicate (*P-value < 0.05, **P-value < 0.01 Statistical analysis performed by two tales student t-test)

was observed for CD69 expression with 8.4% ($P$-value $= 0.001$) and 5.8% ($P$-value $= 0.012$) of cells expressing the marker in GONH$_2$ and GO, respectively. Changes in cell diameter reflect the status of immune cell activation, with larger size corresponding to an active status. The effect of GO and GONH$_2$ on cell size was analyzed using Scepter 2.0. PBMCs were treated with GO and GONH$_2$ at the concentration of 50 μg ml$^{-1}$ for 24 h. Figure 5d shows the cell diameter of lymphocytes, inactivated monocytes (Fig. 5d(i)), and activated monocytes (Fig. 5d(ii)). GONH$_2$ induced higher changes in monocyte diameter compared to the untreated sample. In line with previous results on the action of GONH$_2$, we found $2.65 \times 10^4$ cells with a diameter larger than 11.75 μm, compared to $9.78 \times 10^3$ cells for the control (Fig. 5d (iii)). The effect of GONH$_2$ was studied also by multiplex ELISA on the PBMC supernatants. The secretion of classical Th1/M1 cytokines such as CCL5, IL6, IL1β, and TNFα increased after stimulation with GONH$_2$ but not GO treatment (Fig. 5e; $P$-value $< 0.05$).

All data obtained through the use of classical techniques confirmed the CyTOF main findings, therefore supporting its use as a robust technique for comprehensive analyses of nanomaterial–immune cell interaction.

**Whole-genome expression analysis on T cells and monocytes**. To obtain a higher intensity portray of the interaction between nanomaterials and immune cells, we used the Illumina Beadchip HumanHT-12 v4 genome-wide technology analyzing about 47,000 transcripts in GO- and GONH$_2$-treated T lymphocyte (Jurkat cells) and monocyte (THP1) cell lines as representative of adaptive and innate immune responses, respectively. These cell lines were incubated with GO and GONH$_2$ (50 μg ml$^{-1}$, 24 h) in the same conditions used for the previous CyTOF experiments. To compute the probability of genes being differentially expressed, we used a random variance $t$-test as implemented in BRB Array-Tools (Supplementary Data 1, 2). Results were controlled for false discovery rate (FDR). We confirmed that the functionalization significantly reduced ($P$-value $< 0.001$ and FDR $< 0.05$) the magnitude of the perturbations induced by GO (Venn diagram, Fig. 6a–c). Overall, the number of transcripts modulated by GONH$_2$ was less than one-third of the transcripts altered by GO. Following the treatment with functionalized GONH$_2$, 1163 transcripts were altered in T cells and 977 in monocytes as compared with 4509 transcripts in T cells and 3528 in monocytes in GO-treated samples (Fig. 6; Supplementary Data 1, 2). However, the effect of GONH$_2$ was clearly more specific. Indeed, 2845 transcripts were modulated in both T-cell and monocytes by GO. In contrast, only 390 transcripts were modulated in both treated cells by GONH$_2$ (Fig. 6a). Venn diagrams in Fig. 6b, c describe the different modulation induced by GO and GONH$_2$ in the treated cells.

To provide a functional interpretation of the transcriptional changes, we applied Ingenuity Pathway Analysis (IPA). The most differentially affected canonical pathways in GO- compared to GONH$_2$-treated T cells and monocytes are shown in Fig. 6d while the 20 top canonical pathways are shown in Supplementary Fig. 8. While the perturbations induced by GO reflect the triggering of cytotoxic mechanisms, the changes induced by GONH$_2$ consist of the selective immune activation of T cells and monocytes. Indeed, the canonical pathways most significantly affected by GO are eukaryotic initiation factor 2 (EIF2) signaling, oxidative phosphorylation (OXPHOS), and mTOR signaling, all related to cell metabolism and proliferative function. This effect is visible in both cell types (Fig. 6d). More in detail, protein synthesis, as indicated by the negative $Z$-score of the EI2F pathway, was suppressed by GO treatment, in line with the induction of

apoptotic mechanism showed by the CyTOF and flow cytometry analyses. Conversely, the functionalized GONH$_2$ induced a coordinated induction of immune-activator pathways with limited impact on cell metabolism. Almost all the top 20 canonical pathways modulated by GONH$_2$ are related with immune functions (Supplementary Fig. 8). Such pathways include intracellular signaling implicated in the activation of T cells and in the maturation and activation of monocytes (e.g., interferon signaling, interferon regulatory factor (IRF) activation by pattern recognition receptor (PRR), and antigen presenting and inflammasome pathways) (Fig. 6d; Supplementary Fig. 8). These differences between the two GOs was confirmed using the gene set comparison tool in BRB Array-Tools as a scoring test to assign the functional category definitions according to the Gene Ontology database, with a $P$-value $< 0.005$ (Supplementary Data 3, 4). The perturbation of the OXPHOS pathway found also with IPA highlights the impact of GO on cell metabolism (Supplementary Fig. 9a, b). The modulation of IFN signaling in T cells and DC maturation pathway in monocytes induced by GONH$_2$ are represented in Supplementary Figs. 10, 11. Thus, GO compared to GONH$_2$ induces a stronger alteration of pathways related to cellular replication and metabolism (Supplementary Fig. 8; Supplementary Data 3, 4) with a downregulation of OXPHOS pathway (Supplementary Fig. 9a, b). In summary, GONH$_2$ was confirmed to upregulate immune activation pathways without disrupting intracellular homeostasis. A list of immune-related genes modulated by GONH$_2$ in T cells and monocytes is reported in Fig. 6e. These genes include Th1 chemokines[65] such as *CXCL10* (CXCR3 ligand), *CCL3*, *CCL3L3*, *CCL4L1*, *CCL4L2*, and *CCL5* (CCR5 ligands), pro-inflammatory cytokines such as *TNFα* and *IL1β* (Fig. 6e), and master regulators of the cross-talk between innate and adaptive immune response such as *IRF1* and *STAT1*. To validate the Illumina Beadchip data, we performed real-time PCR with highly specific TaqMan probes. Again, *IRF1*, *CCL3L3*, *IL1B*, and *CCL5* were consistently over-expressed only after GONH$_2$ treatment (Supplementary Fig. 12). Remarkably, these genes (i.e., *CXCR3/CCR5* ligands, and the transcription factors *STAT1* and *IRF1*) are central in the induction of immune-mediated tumor rejection[57, 60, 66] and their overexpression in resected tumors has been associated with favorable prognosis[59, 61, 62]. Such transcripts are upregulated in tumors from patients who are more likely to respond to immunotherapeutic approaches such as IL2[67], vaccine[68], adoptive therapies[58], and checkpoint inhibition[69]. Moreover, the efficacy of cancer immunotherapy relies on the ability to trigger a Th1/M1 anti-tumor response through the induction of the expression of the aforementioned transcripts[54, 57]. Recently, the use of different kinds of nanomaterials as immune modulators for vaccine adjuvant or immunotherapy applications have been described[70–73]. Xu et al. proposed the use of polyethylene glycol and polyethyleneimine functionalized GO as vaccine adjuvant. Functionalized GO was found to promote the maturation of DCs, through the activation of multiple toll-like receptor (TLR) pathways while showing low toxicity[74]. In a similar and extensive way, the current morphological and genomic analysis suggests that GONH$_2$ might enable the initiation and induction of monocyte and DC activation, possibly through TLR/PRR interactions. The results on GO underline how this material is affecting mainly the intracellular metabolic processes such as the OXPHOS and ribosomal activity in both T cells and monocytes in a dose-dependent way. This action could enhance the membrane damage and the reactive oxygen species (ROS) production eventually leading to necrosis[9, 75]. This effect of GO was reported also by other authors for cell lines and bacteria. Akhavan et al.[77] for example, described the interaction of the sharp edges of graphene sheets with the cell walls of bacteria and cell lines,

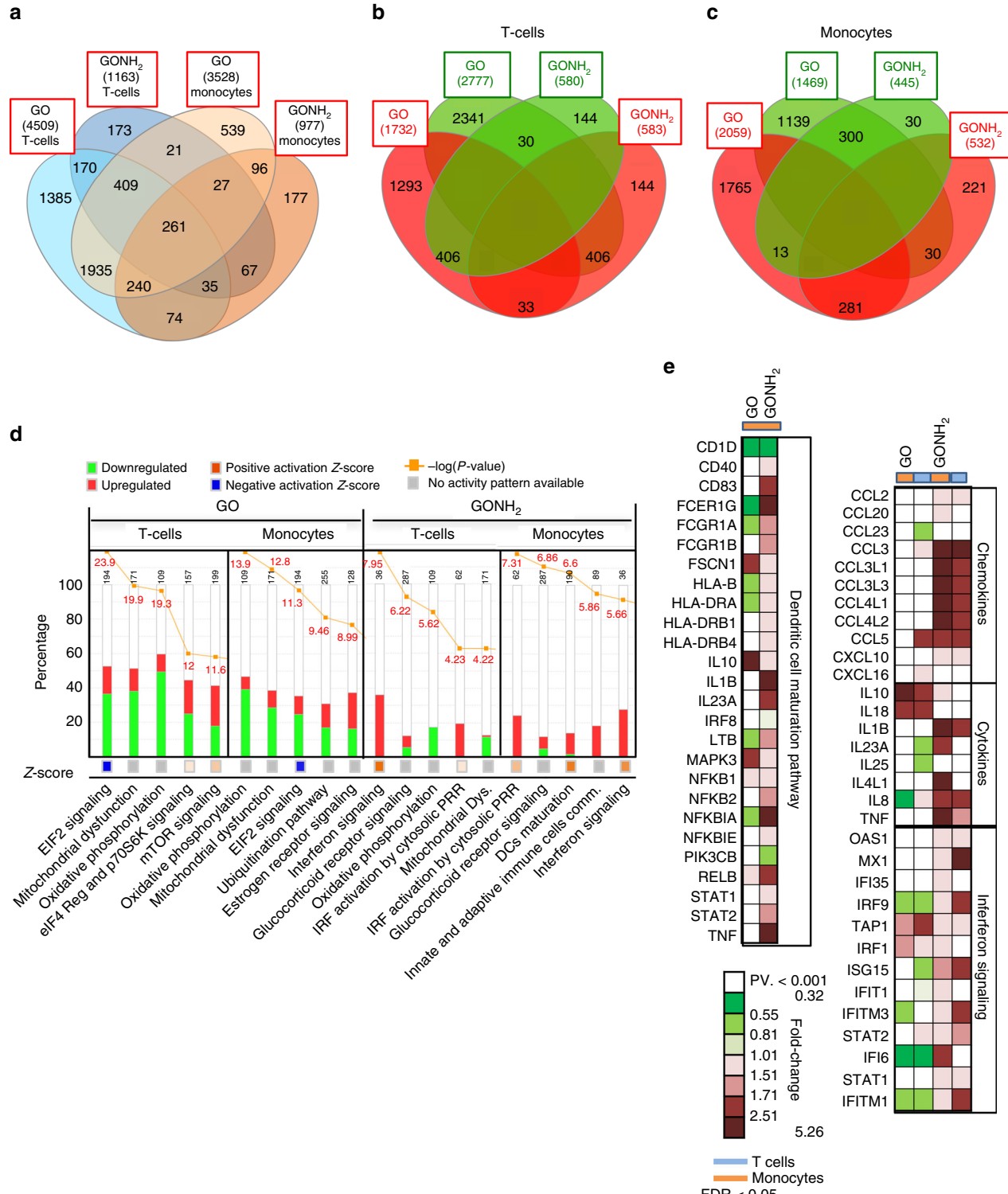

**Fig. 6** Gene expression impact of GO and GONH$_2$ on T- and monocyte cell lines. **a** Venn diagrams for T cells and monocytes reporting the number of transcripts modulated in response to graphene administration. Each Venn diagram is divided into four areas, one for each type of GO and GONH$_2$ in T cells and monocytes. The total number of modulated genes in each area is reported between the parentheses. **b** Venn diagrams of differently expressed transcripts in T cells and **c** in monocytes. Overlapping areas indicate the number of transcripts commonly changed in their expression level between GO and GONH$_2$. Colored circles indicate the number of upregulated (red) and downregulated (green) transcripts with an absolute fold change > 1. **d** Top five first canonical pathways ranking according to significance level (Fisher exact test log (P-value) reported in red) modulated by the GO and GONH$_2$ in T cells and monocytes identified using gene enrichment analysis. The Z-score of each pathway is expressed under each column; the –log(P-value) is reported in red on top of each histogram. **e** Expression heat map of chemokines, cytokines, dendritic cell maturation, and interferon signaling pathway genes (as listed in IPA software) in GONH$_2$- and GO-treated cells vs. Controls; P-value < 0.001 and FDR < 0.05

leading to the generation of ROS and cell wrapping[76]. However, the mechanism associated with cytotoxicity of GO on primary immune cells is likely more complex than in the case of other types of mammalian (i.e., cancer cell lines) and bacterial cells involving a direct impact on cell membrane or generation of ROS, and it warrants future studies. Our data suggests that amino-functionalized GO is likely to facilitate the differentiation of monocytes into monocyte-derived DCs (moDCs). MoDCs pulsed with certain tumor-associated antigens (and eventually prompted with functionalized GO) could expand tumor-specific T-cytotoxic cells to elicit anti-tumor immunity[73, 78, 79].

In conclusion, we propose a high-throughput strategy for the characterization of complex interactions between nanomaterials and the plethora of immune cell populations. The analysis of multicellular systems where cells carry out a diverse array of complex, and specialized functions is still a big challenge[33]. Single-cell mass cytometry enters successfully in this context with its unique capacity of simultaneously resolving a large amount of probes on a per-cell basis at high acquisition rates, thereby providing researchers the ability to phenotypically and functionally profile different cell subpopulations. This technology allows the sophisticated analysis of multiple immune cell interactions with nanomaterials, while overcoming the limitations of spectral overlap present in flow cytometry and revealing all the possible modulations at the single-cell level.

Herein we have focused our studies using CyTOF on the effects of GO, one of the most recent biomedically promising nanomaterials, on primary immune cells. Thanks to this advanced technology, confirmed by classical flow cytometry methods, we report that amino functionalization improves the biocompatibility of GO. Moreover, $GONH_2$ was found to induce a cell-specific activation of T cells, DCs, and monocytes, which were polarized to sustain a M1/Th1 immune response. The positive impact of nanomaterials on specific immune cells can serve as a starting point for the development of nanoscale platforms in medicine as immunotherapeutics, vaccine carrier, and nanoadjuvant tools[27]. Our pilot study paves the way for the future use of single-cell mass cytometry for a deep characterization of immune responses to any type of nanomaterials useful for biomedical applications.

## Methods

**Applied strategy for GO and functionalized $GONH_2$ synthesis.** GO was prepared using a modified Hummers' method previously reported by Ali-Boucetta et al.[80] Briefly, 0.2 g of graphite flakes (Barnwell, UK) was added to 0.1 g of $NaNO_3$ in 4.6 ml of 96% $H_2SO_4$. After obtaining a homogenous dispersion, 0.6 g of $KMnO_4$ was slowly added. The temperature was carefully controlled during the reaction and kept between 98 and 100 °C. The mixture was then diluted with 25 ml of deionized $H_2O$. To reduce residual $KMnO_4$, $MnO_2$, and $Mn_2O_7$ we slowly added a solution of 3% $H_2O_2$. The obtained graphitic oxide suspension was further exfoliated and purified by several centrifugation steps until the pH of the supernatant was around 7. Finally, we extracted and diluted the viscous orange/brown gel-like layer of pure GO using MilliQ water.

2,2′-(ethylenedioxy)bis(ethylamine) (410 µl) was added to a 20 ml of a GO dispersion (1 mg ml⁻¹) in deionized water, and the mixture was stirred for 2 days at room temperature. The solution was then filtered using an Omnipore polytetrafluoroethylene (PTFE) membrane (0.45 µm, Millipore). The solid was dispersed in methanol (100 ml), sonicated for 2 min and filtered again. This procedure was repeated with DMF and methanol. The solid was dispersed in deionized water and dialyzed against deionized water using a dialysis membrane of MWCO 12–14,000 Da.

**Characterizations of the materials.** For TEM characterization, 20 µl of sample (100 µg ml⁻¹) was deposited on a carbon-coated copper grid (Electron Microscopy Services, USA). Excess material was removed by filter paper. Imaging was performed using a FEI Tecnai 12 BioTWIN microscope (Techni, Netherlands) at an acceleration voltage of 100 kV. Images were taken with a Gatan Orius SC1000 CCD camera (GATAN, UK). Lateral size distributions were carried out using ImageJ software after counting the lateral dimension of more than 100 individual GO sheets from several TEM images. For AFM, freshly cleaved mica (Agar Scientific, UK) was used as a substrate. Non-functionalized GO samples required a pre-

coating step of the negatively charged mica surface with 20 µl of 0.01% poly-L-lysine (Sigma-Aldrich, UK). All samples were prepared by depositing 20 µl aliquots of the respective GO dispersions (100 µg ml⁻¹) on the mica substrates and allowing them to adsorb for 2 min. Unbound structures were removed by gentle washing with 2 ml of MilliQ water and samples were left to dry at 37 °C. AFM images were acquired in air using a Multimode 8 atomic force microscope (Bruker, UK) in tapping mode, using an OTESPA tip (Bruker, UK) mounted on a tapping mode silicon cantilever with a typical resonant frequency of 300 kHz. Areas corresponding to 512 × 512 points were scanned at a rate of 1 Hz, using an integral gain of 1 and a proportional gain of 5; amplitude set point values were approximately constant across all measurements. The acquired height images were processed using the Nanoscope Analysis software (Version 1.4, Bruker, UK) in order to assess lateral dimensions and thickness of the GO samples. A drop of the original GO dispersions was placed onto a Tensor 27 FT-IR spectrometer (Bruker, UK) equipped with a 3000 Series High Stability Temperature Controller with RS232 Control (Specac, UK) and a MKII Golden Gate Single Reflection ATR system (Specac, UK) for measurements in ATR mode. The drop was allowed to dry on the plate for 5 min at 60 °C, until a dry powder remained. Transmittance spectra of GO were recorded by acquiring 32 scans between 700 and 4000 cm⁻¹ with a resolution of 4 cm⁻¹. Data processing was completed using OriginPro 8.5.1 software (Origin Lab, USA). For Raman spectroscopy, the samples were prepared for analysis by drop casting 20 µl of sample (100 µg ml⁻¹) dispersion onto a glass slide. The samples were left to dry for at least 2 h at 37 °C. The spectra were collected using a DXR micro-Raman spectrometer (Thermo Scientific, UK) using a $\lambda = 633$ nm laser. The spectra were considered between 500 and 3400 cm⁻¹, enabling visualization of the D and G bands. The spectra were collected at a laser power of 0.4 mW at a magnification lens of ×50 with 25 s exposure time, and averaged over five different locations.

**Immune cell purification and cell culture maintenance.** PBMCs were harvested from ethylenediamine tetraacetic acid (EDTA)-venous blood from informed healthy donors (25–50 years old) using a Ficoll-Paque (GE Healthcare, CA, USA) standard separation protocol. Informed signed consent was obtained from all the donors. The Ethics Committee of the University of Sassari reviewed and approved all the protocols performed. All the experiments were carried out in accordance with the approved guidelines. Jurkat and THP1 cell lines were supplied by the ATCC (American Type Culture Collection) and have been tested for mycoplasma contamination. Jurkat cells, THP1, and PBMCs were daily maintained in RPMI-1640 medium added with FBS 10% and 1% of penicillin/streptomycin solution. At least $1 \times 10^6$ cells for sample in each experiment were used. All the experiments were performed in biological and technical triplicate.

**Flow cytometric strategies for graphene interaction analysis.** To evaluate the cytotoxicity of GO and $GONH_2$, PBMCs were incubated for 24 h at 37 °C with increasing doses of each nanomaterial (5, 25, and 50 µg ml⁻¹). Ethanol was used as a positive control, while samples incubated with medium alone were used as negative controls. Apoptotic and necrotic cells were analyzed with: Annexin-V-FITC, PI, and 7AAD dye (Invitrogen, CA, USA).

To analyze the PBMCs activation after treatment with GO and $GONH_2$, experiments were performed with an intermediate concentration of 50 µg ml⁻¹. After 24 h of incubation, cells were stained to identify immune cell populations and immune activation markers. CD25 and CD69 (APC-conjugated anti-CD25, 2A3 clone; PE-conjugated anti-CD69, L78 clone; BD Bioscience, CA, USA) were used as activation markers. Concanavalin A (ConA, 10 µg ml⁻¹) and lipopolysaccharides (LPS, 2 µg ml⁻¹, Missouri, USA) were used as positive controls (Sigma-Aldrich, Missouri, USA). Staining was performed in the dark for 20 min. Cells were analyzed by flow cytometry (FACS Canto II, BD Bioscience, CA, USA).

**Graphene impact analysis using single-cell mass cytometry.** Single-cell mass cytometry analysis was performed using purified PBMCs obtained as described above. PBMCs were seeded at a concentrations of $3 \times 10^6$ cells per well (six multi-well plates) and treated with GO and $GONH_2$ at the fixed concentration of 50 µg ml⁻¹ for 24 h. After the incubation, cells were harvested and washed with phosphate-buffered saline (PBS). Before the staining, cells were incubated for 5 min with Cisplatin-194Pt to a final concentration of 1 µM. After the incubation, cells were washed with Maxpar Cell Staining Buffer using five times the volume of the starting cell suspension.

Cells were then stained using Maxpar Human Peripheral Blood Phenotyping and Human Intracellular Cytokine I Panel Kits (Fluidigm, CA, USA) following the manufacturer staining protocol for cell surface and cytoplasmic/secreted markers.

Briefly, cells were harvested and resuspended in 50 µl of Maxpar Cell Staining Buffer into 15 ml polystyrene tubes for each sample. The surface marker antibody cocktail (dilution of 1:100 for each antibody) was added to each tube (final volume 100 µl). Samples were mixed and incubated for 30 min at room temperature. After incubation, the samples were washed twice with Maxpar Cell Staining Buffer. Cells were then fixed by adding 1 ml of Maxpar Fix and Perm Buffer to each tube and incubated for 10 min. After incubation, cells were washed twice with Maxpar Fix and Perm Buffer and centrifuged for 5 min at 800×g. Cells were then suspended in 50 µl of Maxpar Fix and Perm Buffer and incubated as described above with

cytoplasmic/secreted antibody cocktail (dilution of 1:100 for each antibody final volume 100 µl). After the incubation, cells were washed twice with Maxpar Cell Staining Buffer and incubated with Cell-ID Intercalator-Ir solution at the final concentration of 125 nM into Maxpar Fix and Perm Buffer for 5 min. Each sample was then washed twice with Maxpar Cell Staining Buffer and suspended with 2 ml of ultrapure water. Before the data acquisition, each sample was filtered into 5 ml round bottom polystyrene tubes with a 30 µm cell strainer cap to remove possible cell clusters or aggregates. Data were analyzed using mass cytometry platform CyTOF2 (Fluidigm Corporation, CA, USA).

**Hemolysis analysis**. Hemolysis test was conducted following previously used protocols[53]. Fresh human whole blood was taken from volunteer healthy donors stabilized with 0.2% EDTA. Informed signed consent was obtained from all the donors. The study was reviewed and approved by the Ethics Committee of the University of Sassari. Serum was removed from blood samples by centrifugation at $200 \times g$ for 5 min. Resulting RBCs were washed five times with sterile isotonic PBS and then diluted 10× with 0.2% EDTA. The hemolytic activity of GO and $GONH_2$ at different concentrations (5, 25, 50, 100 µg ml$^{-1}$) was determined by the incubation of graphenes with the RBC suspension (0.2 ml, $4 \times 10^8$ cells·per ml) in a final volume of 1 ml, completed with PBS. After vortexing, the mixtures were left at room temperature for 2 h, Intact RBCs were removed by centrifugation. A microplate reader (Sunrise, Tecan) measured the absorbance ($A$) of the hemoglobin in the supernatant at 570 nm, with the absorbance at 620 nm as a reference.

**Genomic and cytokines analysis of treated immune cells**. Gene expression analysis was performed as previously described[23, 25]. Briefly, total mRNA from T-cell and monocyte cell lines treated with GO and $GONH_2$ at the concentration of 50 µg ml$^{-1}$, was extracted with TriZol Reagent (Invitrogen, CA, USA) and purified with the RNAeasy Mini Kit (Qiagen, CA, USA). RNA purity was assessed using the Bioanalyzer 2100 (Agilent). Samples with RIN (RNA Integrity Number) < 8 were discarded. About 1 µg of RNA was converted in cRNA and labeled using the Illumina totalPrep RNA Amplification Kit, (Ambion, CA, USA). Biotinylated cRNA was hybridized onto the Illumina HumanHT-12 v4 chip (Illumina, Inc., CA, USA). Probe intensity and gene expression data were generated using the Illumina GenomeStudio software V2011.1 (Gene Expression Module V1.9.0).

The cytokine analysis was performed with MILLIPLEX MAP plex Cytokine Kit (HCYTOMAG-60K, Millipore, MA, USA) (IL1β, TNFα, IL6, Rantes (CCL5)). Cell culture supernatants from PBMCs of at least three experiments were used for the analysis.

**Gene expression data confirmation using real-time PCR**. Total RNA (1 µg) was purified as described above, and reverse transcribed using the superscript IV Reverse Transcription Kit (Invitrogen, CA, USA) following the manufacturer protocol. Real-time PCR reaction was performed on an Applied Biosystems 7300 thermal cycler following the Taqman gene expression assay protocol. Applied Biosystems real-time PCR master mix and the following premade Taqman probes were used: GAPDH Hs99999905_m1 and ACTB Hs99999903_m1 as housekeeping genes, IRF1 Hs00971966_g1, CCL3L3 Hs03407473_uh, CCL5 HS99999048_m1, and IL1B Hs01555410_m1. All experiments were performed in triplicate. Gene expression analysis was computed by the $2^{\Delta\Delta cT}$ method.

**Gating strategy and statistical analysis methods applied**. The analysis of CyTOF data was performed as previously described by Bendall et al.[48] Briefly, normalized, background subtracted FCS files were imported into Cytobank for analysis. Cell events were gated excluding the cell debris, doublets, and dead cells using the Cell-ID Intercalator-Ir and CIS. We defined specific PBMC subsets and subpopulations as reported in Supplementary Fig. 13, in detail: T cells (CD45+ CD3+), T helper (CD45+ CD3+ CD4+), T cytotoxic (CD45+ CD3+ CD8+), T naive (CD45RA+ CD27+ CD38− HLADR−), T effector (CD45RA+ CD27− CD38− HLADR−), T memory (CD45RA− CD27+ CD38− HLADR−), and activated (CD38+ HLADR+), B cells (CD45+ CD19+), B naive (HLADR+ CD27−), B memory (HLADR+ CD27+), plasma B (HLADR− CD38+), NK cells (CD45+ CD3− CD19− CD20− CD14− HLADR− CD38+ CD16+), monocytes (CD45+ CD3− CD19− CD20− CD14+ HLADR+), mDC (CD45+ CD3− CD19− CD20− CD14− HLA- DR+ CD11c+ CD123−), and pDC (CD45+ CD3− CD19− CD20− CD14− HLADR+ CD11c− CD123+). The heat map visualization comparing marker fluorescence of the treated population with mean fluorescent intensity vs. the untreated control was performed with Cytobank. Singlets-gated FCS files were analyzed in Cytobank for spanning tree visualization and comparison. SPADE and viSNE tools were employed. viSNE is a cytometry analysis tool implemented in Cytobank that use t-stochastic neighbor embedding (t-SNE) representing individual cells in a two- or three-dimensional plot, based on their relationships. To construct the SPADE tree and the viSNE map, we used 11 cell surface markers listed as follow: CD3, CD4, CD8a, CD19, CD14, CD16, CD11c, CD123, CD45RA, CD27, and HLADR. Statistical analyses confirming the robustness of the SPADE data (as reported in Supplementary Table 1) were performed using a two-way analysis of variance (ANOVA) and Tukey's multiple comparison test of every algorithm run performed.

Data analysis for flow cytometry data were performed using FACS Diva software (BD Bioscience CA, USA) and FlowJo (LLC, Oregon, USA). Statistical analyses were performed using a two tale Student's $t$-test and one-way ANOVA test. Data with a $P$-value < 0.05 were considered statistically significant. Data are presented as mean ± SD. Multiplex ELISA tests on isolated human primary PMBCs were performed in samples from at least three different donors. Whole-genome expression measurements, and analysis, were performed as previously described[23]. Briefly, whole-genome expression data were extracted and normalized using the Illumina GenomeStudio software V2011.1 (Gene Expression Module V1.9.0). The quality control analysis of the beadchips was performed with the same software. Statistical analysis, visualization of gene expression data, and analysis with GOd were performed using BRBArrayTools (http://linus.nci.nih.gov/BRB-ArrayTools.html). All the genes differentially expressed among the two classes were identified by using a multivariate permutation test with 80% confidence and a false discoveries rates < 5%. Finally, to visualize the expression levels of identified genes for GO and $GONH_2$ in relevant pathway charts, we used IPA (Qiagen, CA, USA).

**Data availability**. The gene expression data sets generated and analyzed during the current study are available in Gene Expression Omnibus (GEO) database (#GSE99929) and as supplementary data set files. Single-cell mass cytometry data and all the other data supporting the findings of this study are available within this article (and its Supplementary Information file), and from the corresponding author upon reasonable request.

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

## Acknowledgements

This work was partly supported by the Fondazione Banco di Sardegna (grant no. 2015.0077 to L.G.D.), the Italian Association against Leukemia (AIL) (early career grant to L.G.D.). This project has received funding from the European Union's Horizon 2020 research and innovation programme under the Marie Sklodowska-Curie grant agreement no. 734381 (CARBO-immap), and the DEPTH project of the European Research Council (grant agreement 322749 to G.C.). This work was partly supported by the Centre National de la Recherche Scientique (CNRS) by the Agence Nationale de la Recherche (ANR) through the LabEx project Chemistry of Complex Systems (ANR-10-LABX-0026_CSC) (to A.B.), and by the International Center for Frontier Research in Chemistry (icFRC). We gratefully acknowledge financial support from EU H2020-Adhoc-2014-20 Graphene Core1 (no. 696656), and from ANR (ANR-15-GRFL-0001-05) and FLAGERA JTC Graphene 2015 (G-IMMUNOMICS project). L.N. and F.A.R. would like to acknowledge the Engineering and Physical Sciences Research Council (EPSRC) Now-Nano DTC and GrapheneNOWNANO CDT programmes at the University of Manchester for their fully funded student scholarships. K.K. would also like to thank the EPSRC Programme Grant 2D-Health (EP/P00119X/1) for partially supporting this research. We thank the staff in the Faculty of Life Sciences EM Facility, Dr. Aleksandr Mironov and Ms. Samantha Forbes, for their expertise and the Wellcome Trust for equipment grant support to the Facility. The University of Manchester Bioimaging Facility microscopes used in this study were purchased with grants from the BBSRC, Wellcome Trust, and the University of Manchester Strategic Fund. We thank Dr. Nigel Hodson for his advice in the use of atomic force microscopy and the EPSRC-funded NEXUS facility for the XPS service.

## Author contributions

L.G.D. with help from A.B., K.K. conceived the idea and supervised the experiments. M.O. implemented the experiments with help from F.S., C.F. and L.N. M.O. analyzed the data with contributions from L.G.D., D.B., W.H., F.A.R. and C.M.-M. F.S., F.M.M. and G. C. participated in the discussion of the project. M.O., L.G.D., D.B., A.B., K.K. wrote the manuscript with contributions from all authors.

## Additional information

**Competing interests:** The authors declare no competing financial interests.

