## [Peer Review File · Nature Communications]

Reviewers' comments:

Reviewer #1 (Remarks to the Author):

The probable toxic effects of graphene oxide on human immune cells were presented. The subject is highly interesting and important in upcoming nanomedicine. Hence, the manuscript is publishable. However, there are some points which should be clarified before final publication, as mentioned below:

1. It was stated that "Raman spectroscopy evidenced the presence of the characteristic D and G bands ... in both GO materials, confirming their graphenaceous structure". But, D and G band alone cannot demonstrate the presence and/or formation of GO. In fact, analyzing the 2D band is highly required to judge about the structure of graphene (its single- and/or multi-layer property). See, for example [CA R B O N 8 1 (2 0 1 5) 1 5 8 -1 6 6] for GO and discuss in this regard.

2. It was mentioned that "the lateral dimensions of both GO and GONH₂ sheets ranged between approximately 50 nm and 1 μm". It is a wide range for a detailed study. In fact, recent studies demonstrated that cyto- and even geno-toxicity of graphene sheets is a size-, concentration- and morphological-dependent effect [Biomaterials 33 (2012) 8017-8025], [J. Mater. Chem., 2012, 22, 20626-20633] and [C A R B O N 5 4 (2 0 1 3) 4 1 9 -4 3 1]. Now, which parameter was mainly involved in the cytotoxic effect of graphene, in this work? Lateral size, concentration, nanoscale thickness, and/or composition? As mentioned by the authors "the role of lateral size and chemical functionalization on the immunological properties of GO is poorly characterized and deserves further investigation". This means that this manuscript is one of the further investigations. Hence, this subject should be discussed in the revised version with more details. In addition, the previous related works should be addressed.

3. The GO materials can hardly be functionalized. In fact, reduced graphene oxide or at least partially reduced graphene oxide is often suitable for functionalization. Hence, for practical applications partially rGO sheets may be more suitable. In addition, I guess that the GONH₂ materials are really partially reduced graphene oxide functionalized with NH₂ (as can be seen by FTIR). This should be clarified in the manuscript. Then it should be clarified that GO and rGO can exhibit different cytotoxic effects, as also reported previously (see. e.g., [ACS Nano VOL. 4 • NO. 10 • 5731-5736 • 2010]).

4. "Revealing the interactions of different GOs with this complex system [I mean body of organisms having the required immune system] remains a challenge", as rightly stated by the authors. There is a recent work concerning the long time effects of GO on some important properties of organisms including hormone secretion of mice polluted with GO and the statistical viability of the next generation [Carbon 95 (2015) 309-317]. This can be helpful for the authors and also the readers.

5. What is the main mechanism involved in the toxic effects of GO and its derivatives? Could the authors comment about it? Some mechanisms have been suggested and approved till now, including, 1) direct contact interaction of extremely sharp edges of graphene with membrane of cells [ACS Nano 2011;5:3693-700] and [ACS Nano VOL. 4 • NO. 10 • 5731-5736 • 2010], 2) ROS generation and 3) wrapping/trapping cells by GO sheets being reduced in a cell media [J. Phys. Chem. B 2011, 115, 6279-6288] and [RSC Adv., 2014, 4, 27213-27223]. Now, what is the mechanism contributed in the effect of GO and GONH₂ on the metabolisms of PBMCs.

Reviewer #2 (Remarks to the Author):

In this manuscript, the authors use the single-cell mass cytometry to analyze the effect of

graphene oxide (GO) and GONH2 on several immune cell populations to determine their activation stages. Their analysis of 30 markers discriminating 15 subpopulations of PBMCs reports on how these subpopulations respond to nanomaterial exposure. They find that amino-functionalization improves the biocompatibility of GO, and that GONH2 induces a cell-specific activation of T cells, dendritic cells and monocytes. The results were corroborated using flow cytometry and qPCR. These efforts are appreciated as studies like this can shed light into the utilization of graphene-based nanomaterials for biomedical applications. However, there are a number of concerns that need to be addressed:

-The authors used the SPADE algorithm to construct the cellular hierarchy in PBMCs. Although this algorithm has been widely used in CyTOF data analysis, the inherited stochasticity in this algorithm makes the results different from run to run. A robustness analysis can make the results presented in this study more reliable. There are also recent robust lineage reconstruction algorithms that can be employed (e.g. ECLAIR; PMID: 27207878).

-Along the same line, using the clustering algorithms like SPADE, the single-cell resolution feature of the study is lost. It is not clear what can we learn about the interactions of the nanomaterial and the immune system at the level of individual cells. Using other dimensionality reduction methods like tSNE can be helpful.

-Previous studies on the applications of graphene-based nanomaterials in biomedicine and their effects on the immune system needs to be further elaborated in the Introduction.

-Does the increased thickness of the functionalized GO have any effect on its utilization in biomedical applications?

Minor points:

- The edges in the SPADE tree in Figs 1 and 24 are not visible.
- Please provide the list of 11 CD markers that were used to generate the trees.
- There are several typos in the manuscript: e.g.

Line 39: grapheme-based

Line 57: Extra "A" at the beginning

Line 248: ares

Line 451: 47.000

Line 806: [as] a positive control.

We have carefully considered the valuable and appropriate suggestions requested by the two referees and incorporated the changes in the revised manuscript. The changes in the revised manuscript are highlighted in yellow.

Point-by-point responses to the referees:

Reviewer #1:

We would like to thank this reviewer for the overall positive comments on our manuscript. He found the article publishable in Nature Communications after addressing his comments:

Comment: The probable toxic effects of graphene oxide on human immune cells were presented. The subject is highly interesting and important in upcoming nanomedicine. Hence, the manuscript is publishable. However, there are some points which should be clarified before final publication, as mentioned below:

1. It was stated that "Raman spectroscopy evidenced the presence of the characteristic D and G bands ... in both GO materials, confirming their graphenaceous structure". But, D and G band alone cannot demonstrate the presence and/or formation of GO. In fact, analyzing the 2D band is highly required to judge about the structure of graphene (its single- and/or multi-layer property). See, for example [CA R B O N 8 1 (2 0 1 5) 1 5 8 –1 6 6] for GO and discuss in this regard.

Response: We agree with the Referee that the analysis of the 2D band – the second order of the D peak – of the Raman spectra is one of the criteria to support the presence of single versus multi-layers of graphene. However, increasing defects on the graphene sheets due to the oxidation of graphite to form GO according to the modified Hummers' method, the structure of the 2D band is lost as per the first stage in the amorphous trajectory of carbon (*Phys. Rev. B61 (2000) 14095*) and is indicative of a loss in the degree of three dimensional ordering (*Carbon 22 (1984) 375*). Going from multi-layered graphene to few-layer GO also indicates that the overall crystallinity and electronic structure of individual sheets is attenuated due to the increased structural defects on the GO sheet. This is clearly evidenced in our data, where the 2D peak is greatly reduced in intensity and broadened, compared to that for graphene. Therefore (and unlike the equivalent with graphene) the analysis of this peak is not a reliable factor to draw conclusions regarding the presence of single-layer of GO (*Solid State Comm. 2007, 143: 47-57*). We used AFM and TEM to determine the single- or multi-layered character of our materials. Furthermore, regarding the graphenaceous nature of our material, we have considered two references for comparison, describing the Raman characterization of GO: a) AIP Advances 2012, 2, 032183; b) Sci. Rep. 2016, 6:19491.

Lastly, we have carefully read the paper cited by the Referee where the 2D band is used to characterise reduced graphene oxide, which is the closest analogue to pristine graphene. This is because there is a degree of restoration of the sp^2 crystal lattice upon reduction, at the same time leading to restoration to some extent of the electrical properties of GO. We have included a few sentences to address the possibility to use the 2D band to determine the presence of single layered sheets for different graphene materials. We have cited the appropriate references related to this.

Comment: 2. It was mentioned that "the lateral dimensions of both GO and GONH2 sheets ranged between approximately 50 nm and 1 μ m". It is a wide range for a detailed study. In fact, recent

studies demonstrated that cyto- and even geno-toxicity of graphene sheets is a size-, concentration- and morphological-dependent effect [Biomaterials 33 (2012) 8017-8025], [J. Mater. Chem., 2012, 22, 20626–20633] and [C A R B O N 5 4 (2 0 1 3) 4 1 9 –4 3 1]. Now, which parameter was mainly involved in the cytotoxic effect of graphene, in this work? Lateral size, concentration, nanoscale thickness, and/or composition? As mentioned by the authors "the role of lateral size and chemical functionalization on the immunological proprieties of GO is poorly characterized and deserves further investigation". This means that this manuscript is one of the further investigations. Hence, this subject should be discussed in the revised version with more details. In addition, the previous related works should be addressed.

Response: We would like to thank the Referee for addressing the important issue of the different parameters that need to be considered to show the impact of GO on the ensuing cytotoxicity. The Referee is right about the large distribution of the lateral size. The exfoliation of graphite to yield GO flake forms of which the lateral dimensions can span several orders of magnitude, constituting a log-normal distribution (*J Phys Condens Matter. 2015, 27:013002*). In this study, we have reported the range between the minimum and maximum size of the detected graphene flakes. However, the majority of the GO flakes in our sample preparations have consistently lateral dimensions between 50 and 300 nm. We have now presented the size distribution in a different way, showing average lateral size including the standard deviation, and describing in the text the percentages of the GO around that average. For example, 73% of GO and 62% of GONH₂ flakes have lateral dimensions between 50 and 300 nm, respectively. We have also considered the papers suggested by the Referee and better discussed the parameters mentioned above.

Comment: 3. *The GO materials can hardly be functionalized. In fact, reduced graphene oxide or at least partially reduced graphene oxide is often suitable for functionalization. Hence, for practical applications partially rGO sheets may be more suitable. In addition, I guess that the GONH₂ materials are really partially reduced graphene oxide functionalized with NH₂ (as can be seen by FTIR). This should be clarified in the manuscript. Then it should be clarified that GO and rGO can exhibit different cytotoxic effects, as also reported previously (see. e.g., [ACS Nano VOL. 4 ▪ NO. 10 ▪ 5731–5736 ▪ 2010]).*

Response: We would like to thank the referee for this comment as it allows us to discuss the important issue related to the reactivity of graphene materials. We have a long term experience with the functionalisation of carbon nanomaterials, and we are highly concerned by the reactivity of graphene oxide as in the literature there are many flaws. Due to the presence of many oxygenated groups present on its surface, GO can be easily functionalised by various reactions such as epoxide opening, silanisation, and isocyanate derivatisation, among others (Chem. Rev. 2012, 112, 6027). We recently elucidated the chemical reactivity of GO towards amines by solid-state NMR (Ref. 30, Vacchi et al. Nanoscale 2016). In the case of reduced GO, as there are less functional groups, the number of organic reactions that can be applied is reduced. For some applications, in particular in nanomedicine, the lower water dispersibility of reduced GO can be a limitation and GO may be preferable. The GONH₂ was prepared according to the protocol described in Ref. 30. In that article, we demonstrated by XPS that the GONH₂ was not reduced as the C/O ratio only slightly increased from 2.24 to 2.62 after functionalization. This little increase in overall C/O ratio was explained by the introduction of the triethylene glycol chain, which contains more carbon than oxygen atoms. We

have better clarified this point in the manuscript and stated that the chemical functionalisation through epoxide ring opening does not reduce the material.

Comment: 4. "Revealing the interactions of different GOs with this complex system [I mean body of organisms having the required immune system] remains a challenge", as rightly stated by the authors. There is a recent work concerning the long-time effects of GO on some important properties of organisms including hormone secretion of mice polluted with GO and the statistical viability of the next generation [Carbon 95 (2015) 309-317]. This can be helpful for the authors and also the readers.

Response: We have discussed the results suggested by the referee and cited the related reference.

Comment: 5. What is the main mechanism involved in the toxic effects of GO and its derivatives? Could the authors comment about it? Some mechanisms have been suggested and approved till now, including, 1) direct contact interaction of extremely sharp edges of graphene with membrane of cells [ACS Nano 2011;5:3693-700] and [ACS Nano VOL. 4 ▪ NO. 10 ▪ 5731–5736 ▪ 2010], 2) ROS generation and 3) wrapping/trapping cells by GO sheets being reduced in a cell media [J. Phys. Chem. B 2011, 115, 6279–6288] and [RSC Adv., 2014, 4, 27213–27223]. Now, what is the mechanism contributed in the effect of GO and GONH₂ on the metabolisms of PBMCs.

Response: As suggested we have now discussed the mechanism involved on the impact of GO on certain cell populations, and cited relevant references.

Reviewer #2:

We thank this reviewer for the constructive comments and for pointing out the interest in our manuscript suggesting its publication in Nature Communications after the addressing of her/his comments.

Comment: 1. The authors used the SPADE algorithm to construct the cellular hierarchy in PBMCs. Although this algorithm has been widely used in CyTOF data analysis, the inherited stochasticity in this algorithm makes the results different from run to run. A robustness analysis can make the results presented in this study more reliable. There are also recent robust lineage reconstruction algorithms that can be employed (e.g. ECLAIR; PMID: 27207878).

Response: We thank the referee for his/her comment. As stated by the referee, SPADE is still the most used algorithm to assemble the cellular hierarchy as recently reported by Greenplate AR et al. European Journal of Cancer 61 (2016) 77e84. Moreover, the use of SPADE for PBMC phenotyping is suggested by the FLUDIGM corporation in the technical data sheets. However, the referee is correct in his/her comment about the stochastic nature of the algorithm. For this reason, we performed three independent SPADE analyses of our data to prove the robustness of the derived conclusions. In **Table S1** we report the different event counts for the main immune cell subpopulations. The comparison of the estimate of the cell population size in the different analyses support the statistical significance and the robustness of our conclusions.

As suggested by the referee we also set out to use the ECLAIR algorithm. To this end, we downloaded the software. However, we soon realized that the software is not user friendly. There is no simple user interface and the algorithm can only be operated by inserting python instructions in the command line. We also involved a python expert collaborator but in the absence of any documentation we were not even able to identify the data input format. We did also contact the developers that unfortunately claimed that they were not any longer responsible for supporting the software tool.

To address the referee comment on the stochastic nature of the SPADE approach, we added the following paragraph at pg 6:

These trees provide a convenient approach to map complex n-dimensional relationships into a representative 2D structure⁴¹.

However, it is well known that the SPADE algorithm is inherently stochastic and the estimate of the cell populations differ in different repeat of the analyses. This limit of SPADE is also supported by the continues born of new algorithms⁴². To corroborate the robustness of our conclusions, the SPADE analysis was performed three times. As reported in Supporting Table S1, the events count of the main immune subpopulations are similar in the different algorithm runs, confirming the robustness of the SPADE data analysis.

In materials and methods, we added the following sentences:

Statistical analyses confirming the robustness of the SPADE data (as reported in Supporting Table S1) were performed using a two-way ANOVA and Tukey's multiple comparison test of every algorithm run performed.

Comment: 2. Along the same line, using the clustering algorithms like SPADE, the single-cell resolution feature of the study is lost. It is not clear what can we learn about the interactions of the nanomaterial and the immune system at the level of individual cells. Using other dimensionality reduction methods like tSNE can be helpful.

Response: We agree with this comment and we thank the referee for the suggestion to better explore our data at single cell resolution by using tSNE analysis. To address this comment, we added a few sentences at page 7 and 8 and included a new figure (**Figure 4**) and new Supporting Figures to strengthen the results:

However, the SPADE visualization fails to preserve the single-cell resolution of the mass cytometry data. For this reason, we applied a second dimensionality reduction method called viSNE. viSNE, is a computational approach suitable for the visualization of high dimensional data with single-cell resolution⁵⁸. By this approach, immune cell phenotypes are projected onto a biaxial plot space according to the similarity of their multidimensional phenotypic expression vector. Thus viSNE cluster the single cell events into populations according to the 11 protein expression readouts used in the analysis (**Figure 4**). The viSNE analysis accurately identified helper and CTL T cells, B cells, monocytes and NK cells (**Figure S3, S4, S5 and S6**). The naïve, memory and activated T cell subpopulations and the B cell subpopulations were also identified (**Figure S3, S4 and S5**).

We further exploited the viSNE analysis to investigate the single-cell cytokine profile in response to GO and GONH₂ treatment. This analysis confirmed the subpopulations and cytokine expression profiles obtained by the SPADE approach and supports the main conclusion that the amino functionalization of GO significantly increases cell biocompatibility and polarizes a specific cell activation toward a T helper-1/M1 immune response not affecting the B cell response. Instead, GO incubation caused an increase in B cell count correlating with an increase in IL-2 secretion mostly by the plasma B cells and a reduction of monocytes (**Figure S5** and **S6**). The single-cell resolution obtained with the viSNE analysis evidenced a heterogeneity in the cytokine expression profile within the same subpopulation (TNF α , IL-6, IL-5, IL-4, and IFN γ) (**Figure 4**) revealing a heterogeneous response after GO and GONH₂ treatment.

In materials and methods, we added the following sentences:

For spanning tree visualization and comparison, singlet-gated FCS files were analyzed in Cytobank using the SPADE tool comparing the median fluorescence intensity and viSNE a cytometry analysis tool that employs t-stochastic neighbor embedding (t-SNE) in mapping individual cells in a two or three-dimensional map based on their high dimensional relationships. To construct the SPADE tree and the viSNE map, we used 11 cell surface markers listed as follow: CD3, CD4, CD8a, CD19, CD14, CD16, CD11c, CD123, CD45RA, CD27 and HLADR.

Comment: 3. Previous studies on the applications of graphene-based nanomaterials in biomedicine and their effects on the immune system needs to be further elaborated in the Introduction.

Response: To address this comment we added the following text at page 3 of the introduction:

Thus, graphene can be rich in functional groups such as epoxy and hydroxyl groups which facilitate its surface modifications increasing its biocompatibility. Very recently, GO have been investigated in a growing number of medical applications, such as drug delivery, diagnostics, tissue engineering and gene transfection, all with the final aim to use it as a therapeutic nanomaterial^{12,13}. However, one of the main concerns in using graphene in nanomedicine is its biocompatibility. Similarly, to many other nanomaterials, it is necessary to carefully address its impact on cell viability. On the other hand, specific toxic effects of graphene on cancer cells support a good use in nanomedicine. Indeed, many reports have shown that graphene anti-tumor properties could be exploited in future therapeutic applications^{14,15}, for example as an inhibitor of cancer cell metastasis¹⁶ or as a passive tumor cell killer in leukemia¹⁷. In this context, almost all suggested applications are based on intravenous injection of graphene, and hence involve an interaction with the immune system. As such, the evaluation of the immune impact of the nanoparticles is an essential prerequisite.

As discussed above the effects played by physicochemical characteristics of nanomaterials in terms of lateral dimension, functionalization, and purity are still under discussion. Recent studies have shown that a lateral dimensions lower than 1 μm seems to trigger a higher activation of immune cells while a higher lateral dimension increased biocompatibility¹⁰. In a similar way the chemical modifications of graphene can significantly affect the impact of these nanoparticles on the immune system. It is already reported that functionalization can reduce the toxicity by changing the ability of

graphene to modulate the immune response. The possibility to rationally design graphene materials with different physicochemical characteristics could expand further their application in medicine.

Comment: 4. Does the increased thickness of the functionalized GO have any effect on its utilization in biomedical applications?

Response: We would like to thank the referee for this comment as it allows us to discuss this important issue. The impact of the thickness of GO in a biological environment is still a relatively unexplored issue. The first study that attempted to address this point was recently reported by some of us (*Jasim et al. Applied Materials Today 4 (2016) 24–30*). Our results showed that a significantly larger fraction of the thicker (more than 20nm thick on average) functionalized GO sheets (47.5% of injected dose) remained within the body of living animals 24 h after intravenous administration, residing mainly in the spleen and liver. The thinner (very similar to the amino-functionalized GO used in this study) sheets were predominantly (76.9% of injected dose) excreted through the glomerular filter into the urine. However, it would not be accurate to generalize at this stage since this was the first-ever study to examine the effect of thickness. Until further studies are performed interrogating the effect of GO flake thickness on interaction with different biological matter (cells, tissues, blood, etc) it would be very premature to reach conclusions. In relation to the present study, we have been consistently reporting that chemical functionalization of single or double layered (i.e. very thin) GO leads inevitably to an increase in flake thickness (to about five GO layers). Based on the in vivo data mentioned above, such minute increases in GO sheet thickness do not seem to affect the biological properties or biocompatibility of the material.

Minor points:

- The edges in the SPADE tree in Figs 1 and 24 are not visible.
- Please provide the list of 11 CD markers that were used to generate the trees.
- There are several typos in the manuscript: e.g.

Line 39: grapheme-based

Line 57: Extra "A" at the beginning

Line 248: ares

Line 451: 47.000

Line 806: [as] a positive control.

We thank the referee for pointing out these minor points and typos that we carefully addressed into the manuscript.

Reviewers' comments:

Reviewer #1 (Remarks to the Author):

The authors tried to revise the manuscript based on the comments. But, there are still some points which required further clarifications, as mentioned below:

Concerning Comment #2, the authors tried to further highlight the role of lateral dimension. In addition they mentioned that further discussion was presented based on the related literature ("We have also considered the papers suggested by the Referee and better discussed the parameters mentioned above"). But, I could not find any supported discussion in this regard.

Concerning Comment #3, I prefer to work with O/C ratio. It was stated that O/C ratio decreased from 0.44 into 0.38 which is a considerable reduction for showing partial reduction (please see related literature in this regard). For example, reduction by hydrazine typically results in O/C ratios of ~0.2 (see, for example, [C A R B O N 6 6 (2 0 1 4) 3 9 5 -4 0 6]). The authors should further discuss in this regard using suitable supports in the revised manuscript. In addition, as mentioned previously, they should discuss about the different behavior of GO and rGO in toxicity and compare with with previous reports.

Concerning Comment #4, it was stated that "We have discussed the results suggested by the referee and cited the related reference. ". But, I could not find the discussion. Please clarify.

Concerning Comment #5, it was stated that "As suggested we have now discussed the mechanism involved on the impact of GO on certain cell populations, and cited relevant references.". Unfortunately, once again, I could not find any supported discussion in this regard.

After clarifying these points the manuscript can be reconsidered for publication.

Reviewer #2 (Remarks to the Author):

The authors addressed all my concerns in the revised manuscript.

We have carefully considered the new comments of the first referee and incorporated the changes in the revised manuscript. The changes in the revised manuscript are highlighted in yellow.

Point-by-point responses to the referee:

Reviewer #1:

The authors tried to revise the manuscript based on the comments. But, there are still some points which required further clarifications, as mentioned below:

Response: We thank again this referee for the constructive comments and for writing "after clarifying these points the manuscript can be reconsidered for publication".

Concerning Comment #2, the authors tried to further highlight the role of lateral dimension. In addition, they mentioned that further discussion was presented based on the related literature ("We have also considered the papers suggested by the Referee and better discussed the parameters mentioned above"). But, I could not find any supported discussion in this regard.

Response: We sincerely apologize for having overlooked this part relative to the discussion of previous papers describing the parameters influencing the cytotoxic effects of GO. In the revised version, we have now added the following sentences supported by the corresponding references:

"Similarly, the cyto- and genotoxicity of reduced GO (rGO) sheets on human mesenchymal stem cells were found to depend on the lateral dimensions of the materials, ultra-small sheets being more toxic¹⁸. Interestingly, in a separate study by the same group, the use of GO sheets as opposed to rGO sheets of lateral dimensions $\sim 2 \mu\text{m}$ was associated with a modest reduction in cytotoxicity ($\sim 20\%$) towards this same cell type¹⁹. Studies have also shown that the aspect ratio of the graphene sheets is also an important factor to consider. For instance, reduced GO (rGO) affects cell viability only at very high concentration (i.e. $100 \mu\text{g}\cdot\text{mL}^{-1}$), while single-layer rGO nanoribbons display significant cytotoxic effects at $10 \mu\text{g}\cdot\text{mL}^{-1}$. Intriguingly, this has also a direct impact on the antibacterial activity of GO and rGO (i.e. rGO is therefore more efficient as bactericide than GO)²⁰."

Concerning Comment #3, I prefer to work with O/C ratio. It was stated that O/C ratio decreased from 0.44 into 0.38 which is a considerable reduction for showing partial reduction (please see related literature in this regard). For example, reduction by hydrazine typically results in O/C ratios of ~ 0.2 (see, for example, [C A R B O N 6 6 (2 0 1 4) 3 9 5 –4 0 6]). The authors should further discuss in this regard using suitable supports in the revised manuscript. In addition, as mentioned previously, they should discuss about the different behaviour of GO and rGO in toxicity and compare with with previous reports.

Response: We thank the referee for the input on this point. However, we consider a difference in the O/C ratio of 0.06, as not significant to account for reduction of GO. Indeed, the referee is mentioning a typical O/C ratio of 0.2 when GO is reduced by hydrazine. We would like to point out that hydrazine is not considered a molecule of the family of the amines. This reagent is well known as a reductant able to eliminate

oxygen from oxygen rich compounds. The mechanism of reduction of GO, by hydrazine, has been described in different papers and clearly illustrated in a recent review article (Chua et al. Chem. Soc. Rev. 2014, see scheme 3). This mechanism is different from a simple epoxide ring opening reactions occurring in the presence of aliphatic amines. We have strong evidence that TEG-diamine does not reduce GO as we could see from the XPS analysis. We did not report in this paper the XPS analysis, but we have currently a manuscript, in preparation, where we used the epoxide ring opening to functionalize GO for another study. The same strategy has been used also our recent paper by Vacchi et al. Nanoscale 2016, and XPS proved that there is no significant reduction of C-O energy band as in the case of a reduction using appropriate reagents (i.e. hydrazine). Ethylene diamine was also reported to reduce GO, but the mechanism goes through a cyclization step that is not possible in the case of TEG-diamine. We have discussed these points in the revised version adding the following sentence.

"It was reported that a strong reduction process results in an increased I(D)/I(G) ratio due to the predominance of small sp² carbon domains in the graphene lattice⁴⁴. These results are consistent with the maintenance of the oxidation degree of GO after functionalization, which was demonstrated in a previous study by XPS analysis of the C-O binding energy peak, before and after functionalization with TEG diamine³⁰. We found that the O/C ratio decreased from 0.44 to 0.38 after GO covalent modification. It was reported that ethylenediamine is able to reduce GO, but the mechanism involves the formation of a 5-membered ring that is not possible using TEG diamine⁴⁵. Stronger agents like hydrazine or plant extracts are necessary to achieve an efficient reduction of GO^{45, 46}."

Concerning Comment #4, it was stated that "We have discussed the results suggested by the referee and cited the related reference. ". But, I could not find the discussion. Please clarify.

Response: We apologize again for having missed the discussion of the important point suggested by the referee. We have now added a short discussion inserting the following sentence into the introduction section:

"Recently, interesting studies analyzed also the long-time effects of GO and rGO on reproduction capability of mice. Significant damaging effects on spermatozoa were observed at very high injected doses, in addition to the disturbance to hormone secretion patterns, as well as pregnant functionalities in females and consequently the viability of the next generation^{21, 22}."

Concerning Comment #5, it was stated that "As suggested we have now discussed the mechanism involved on the impact of GO on certain cell populations, and cited relevant references." Unfortunately, once again, I could not find any supported discussion in this regard.

Response: We have made an attempt to discuss the mechanism involved concerning the impact of GO, in comparison to related data available in the literature. We have added the following sentences and relevant references:

"The results on GO underline how this material is affecting mainly the intracellular metabolic processes affecting the OXPHOS and ribosomal activity in both T cells and monocytes in a dose dependent way. This action could enhance the membrane damage and the reactive oxygen species (ROS) production eventually leading to necrosis as we reported for primary macrophages¹⁰. This effect of GO was reported

also by other authors for cell lines and bacteria. Akhavan et al. for example, described the interaction of the sharp edges of graphene sheets with the cell walls of bacteria and cell lines, leading to the generation of ROS and the cell wrapping^{80, 81}. However, the mechanism leading to cytotoxicity of GO on primary immune cells is likely more complex than in the case of other types of mammalian (i.e. cancer cell lines) and bacterial cells involving a direct impact on cell membrane or generation of ROS, and it warrants future studies."

"High amounts of necrotic cells were detected, suggesting a possible direct effect of GO on the cell membrane that leads to extensive damage. These findings were in agreement with a previous work in which we disclosed the "mask effect" of GO¹⁰."

We hope to have carefully addressed the points raised by the referee and that the revised version of the manuscript is now suitable for publication.

REVIEWERS' COMMENTS:

Reviewer #1 (Remarks to the Author):

The authors revised the manuscript based on the comments and now the manuscript is publishable.